# A Genetic Study of Spillovers in the Bean Common Mosaic Subgroup of Potyviruses

**DOI:** 10.3390/v16091351

**Published:** 2024-08-23

**Authors:** Mohammad Hajizadeh, Karima Ben Mansour, Adrian J. Gibbs

**Affiliations:** 1Department of Plant Protection, Faculty of Agriculture, University of Kurdistan, Sanandaj 66177-15175, Iran; 2Ecology, Diagnostics and Genetic Resources of Agriculturally Important Viruses, Fungi and Phytoplasmas, Crop Research Institute, Drnovská 507, 161 06 Prague, Czech Republic; karima.ben_mansour@vurv.cz; 3Department of Plant Protection, Faculty of Agrobiology, Food and Natural Resources, Czech University of Life Sciences Prague, Kamýcká 129, 165 00 Prague, Czech Republic; 4Emeritus Faculty, Australian National University, Canberra, ACT 2601, Australia

**Keywords:** potyvirus, spillovers, emergences, phylogenetics, population genetics, codon selection, protein modelling

## Abstract

Nine viruses of the bean common mosaic virus subgroup of potyviruses are major international crop pathogens, but their phylogenetically closest relatives from non-crop plants have mostly been found only in SE Asia and Oceania, which is thus likely to be their “centre of emergence”. We have compared over 700 of the complete genomic ORFs of the crop pandemic and the non-crop viruses in various ways. Only one-third of crop virus genomes are non-recombinant, but more than half the non-crop virus genomes are. Four of the viruses were from crops domesticated in the Old World (Africa to SE Asia), and the other five were from New World crops. There was a temporal signal in only three of the crop virus datasets, but it confirmed that the most recent common ancestors of all the crop viruses were before inter-continental marine trade started after 1492 CE, whereas all the crown clusters of the phylogenies are from after that date. The non-crop virus datasets are genetically more diverse than those of the crop viruses, and Tajima’s D analyses showed that their populations were contracting, and only one of the crop viruses had a significantly expanding population. dN/dS analyses showed that most of the genes and codons in all the viruses were under significant negative selection, and the few that were under significant positive selection were mostly in the PIPO-encoding region of the P3 protein, or the PIPO protein itself. Interestingly, more positively selected codons were found in non-crop than in crop viruses, and, as the hosts of the former were taxonomically more diverse than the latter, this may indicate that the positively selected codons are involved in host range determination; AlphaFold3 modelling was used to investigate this possibility.

## 1. Introduction

The appearance in the second half of 2019 of a major viral pandemic on the human population has recharged interest in how viruses “emerge” or “spillover” from the wild and cause pandemics. Some plant pandemics are devastating, and many are less obvious but nonetheless damaging [1]. However, the sources of most viral pandemics are usually unknown and often controversial [2]. The bean common mosaic virus (BCMV) subgroup of potyviruses is of interest as it contains many viruses of non-crop species but also many well-known crop pandemic viruses, which might have resulted from spillovers.

Viruses of the BCMV subgroup were first recognised more than a century ago [3] when a mosaic disease of *Phaseolus vulgaris*, the common bean, was first described in Russia by Iwanowski in 1899, then in the eastern U.S.A. by Stewart in 1917 [4], and in more detail by Pierce in 1934 [5]. The disease was shown to be seed-borne, sap-transmitted but with difficulty [6], and also some isolates were transmitted by several species of probing aphids [7]. Host range tests indicated that there were probably several different viruses causing the bean mosaic pandemic, as well as similar diseases of other cultivated legumes such as soybean, adzuki, etc. Serological tests [8] failed to distinguish between these viruses as, it was later found, the principal antigenic sites on the virions of potyviruses are poor immunogens and unstable. However, in 1959, Brandes showed that all viruses with rod-shaped or filamentous virions could be placed into usefully predictive groups by the length and morphology of their virions [9,10], and these criteria showed that most of the different viruses causing bean mosaic pandemics were placed in a group with potato virus Y, which provided the name “potyvirus” [11]. Finally, peptide analysis [12] and then gene sequencing [13,14,15] showed that many of the seed-borne potyviruses of common bean and other legumes formed several clearly defined species in a subgroup of the potyviruses, which was named after bean common mosaic virus (BCMV) by Dijkstra in 1992, who included viruses from both leguminous and non-leguminous host plants [16].

Gibbs et al. [17] analysed the coat protein genes of potyviruses isolated in Australia and found that 17 Australian potyviruses isolated from various crops were from 13 different potyvirus lineages and were closely related to them, whereas another 18 Australian potyviruses found in native plants but 14 were from the BCMV subgroup and related more distantly to viruses of the BCMV subgroup isolated from uncultivated plants or indigenous crops in SE Asia. They concluded that most crop potyviruses are recent migrants to Australia, whereas the BCMV subgroup of potyviruses had probably first diversified in SE Asia and spread to Australia sometime in the past. Thus, the BCMV subgroup viruses present a rare, perhaps unique, opportunity to make genetic comparisons of several related crop viruses with their possible source populations, and this may provide new ideas on the features that permit or promote spillovers.

The genomes of all potyviruses [18] are single molecules of ss-RNA and have short terminal non-coding regions. Each genome mostly encodes a single polyprotein that, after transcription, is hydrolysed by the three proteases it includes [19] to produce ten proteins. The proteins are the N-terminal P1 protein (a self-hydrolysing serine protease) and the HC-pro (a helper component and self-hydrolysing cysteine protease). The remaining polyprotein is hydrolysed at seven sites by NIa-pro (the eighth protein) to produce seven proteins: 6K1, CI (the cytoplasmic inclusion protein), 6K2, NIa-VPg (the genome 5′terminal capping protein), NIa-pro, NIb (the RNA-dependent RNA polymerase), and, at the 3′ end, CP (the coat protein gene). In the middle of the P3 genes of all potyviruses is a conserved motif of six adenine residues [20] that produces transcriptional slippage [20] in a small proportion of transcriptions, and these translate to produce the eleventh protein, a shortened transcript with a novel 3′ terminus called P3N-PIPO. In viruses of the sweet potato feathery mottle virus subgroup, another similar overlapping gene of the P1 protein produces a twelfth protein [21,22]. A considerable body of research has aimed to determine the functions of the various proteins; however, it is clear that most act cooperatively in a complex biochemical life cycle [23,24,25,26,27].

The BCMV subgroup is now represented in the international GenBank database by over 800 full-length genomic sequences, and this allows for a range of genetic comparisons to be made. Several of these viruses were first found in crop plants and are represented in the GenBank database by more sequences than others, probably because crop pathogens are of more interest to plant pathologists than non-crop pathogens, so we used a count of the number of sequences in GenBank to sort the BCMV subgroup sequences into the “crop” and “non-crop” or “outgroup” viruses. We then compared the gene sequences of the crop viruses with those of the phylogenetically closest non-crop viruses in various ways, looking for differences that may have arisen during spillovers and/or adaptation to crops.

## 2. Materials and Methods

All the full-length genomes of viruses of the BCMV subgroup were downloaded from GenBank. To obtain these, the genomic sequences of the 22 virus isolates contributing to the BCMV group in Figure 3 of Ref. [28] were each used for a Discontiguous Mega BLAST search of the GenBank database, each collecting the nearest 250 sequences. The 5500 resulting sequences were pooled, the duplicates were removed, and around 800 unique sequences remained, of which 731 were complete, or nearly so. Their Accession numbers are listed in Appendix A.

The 731 unique sequences were aligned by MAFFT using default parameters and trimmed using BioEdit 7.2.5 [29] to obtain the main open reading frames (ORFs) used for all analyses. ORFs were aligned using MAFFT [30] with default parameters and, after trimming and degapping, were aligned by PAL2NAL [31]. The aligned sequences were tested for the presence of phylogenetic anomalies probably resulting from recombination using all the options in RDP v.5.5 [32], with default parameters, namely that an anomaly was detected by five or more methods with an average chance probability of <10^−5^ [33,34]. Appendix A is a histogram showing the vital statistics of the 11,001 nt (3667 codon) alignment of 731 sequences we analysed. The NJ tree option in ClustalX [35] was used to calculate, with default parameters, a phylogeny of the ORFs (Appendix A). All the sequence names were checked phylogenetically, and several sequences with unique names were found to be isolates of others, notably isolates of BCMV, WMV, or ZYMV, and were therefore grouped with them.

The best-fit substitution model for the ORFs was assessed using MEGA11 [36] and found to be GTR + γ4 + I, and LG + γ4 + I for their encoded amino acid sequences. Maximum likelihood (ML) phylogenies were calculated using PhyML3.0 or MEGA11. Phylogenies were dated using IQ-TREE 2.3.4-Windows [37]. The statistical support for nodes of ML trees was assessed using the SH option in PhyML [38]. Phylogenies were drawn using Figtree Version 1.4.4 “http://tree.bio.ed.ac.uk/software/FigTree/ (accessed on 30 November 2023) and a commercial graphics package.

The ORFs of all 248 non-recombinant (n-rec) genomes were used to generate a ML phylogeny using MEGA11, and this was converted to a pairwise patristic distance matrix using PATRISTIC [39]. The matrix was interrogated in MS Excel to identify the 15 sequences with the smallest average patristic distance for each of the crop virus clusters. We call these clusters “outgroup” clusters as some included crop sequences but no more than two sequences were chosen for each set from each of the other crop virus clusters. PATRISTIC was also used for comparing the phylogenies calculated from alignments of nucleotide sequences and the amino acid sequences they encoded.

The program DnaSP v.6.10.01 [40] was used to analyse the genetic differences between each of the crop virus populations and its 15 outgroup sequence populations (Appendix A). Estimates were calculated for the average pairwise nucleotide diversity (π), number of segregation sites (S), mean non-synonymous substitutions per non-synonymous site (dN), mean synonymous substitutions per non-synonymous site (dS), and the ratio of non-synonymous nucleotide diversity to synonymous nucleotide diversity (dN/dS). It was concluded that genes were under positive, neutral, or negative selection when their dN/dS ratios were >1, =1, and <1, respectively [34]. In addition, Tajima’s D genetic tests of neutrality were used to determine whether the populations had a greater or smaller diversity than expected. The FUBAR (Fast Unconstrained Bayesian AppRoximation) method [41] implemented in the online Datamonkey server (https://www.datamonkey.org/ (accessed on 30 November 2023)) was used to find evidence of “significant pervasive selection pressures”, both positive and negative, on individual codons in the genes.

The AlphaFold3 server [42] was used to model the amino acid sequences of individual genes and combinations of the genes of the peanut mottle virus (PeMoV) and the dasheen mosaic virus (DashMV). The models obtained from AlphaFold 3 were investigated using Visual Studio Code 1.90. The Protoparam tool of Expasy [43] was used to estimate the pI and hydropathicity of proteins.

## 3. Results

### 3.1. The Data

All the full-length genomic sequences of viruses of the BCMV subgroup were downloaded (February 2023) from GenBank, and 731 unique sequences were found (Appendix A and Appendix A). Nine viruses (Figure 1) were found to be represented by the complete genomes of 15 or more isolates, which indicates that these viruses had attracted the attention of plant pathologists. They are the bean common necrosis virus (BCMNV), bean common mosaic virus (BCMV), cowpea aphid-borne mosaic virus (CpAbMV), dasheen mosaic virus (DashMV), East Asia passiflora virus (EAPV), peanut mottle virus (PeMoV), soybean mosaic virus (SbMV), watermelon mosaic virus (WMV), and zucchini yellow mosaic virus (ZYMV). We call these the “crop viruses”. The other 37 viruses were represented by the genomic sequences of eight or fewer isolates (see Figure 1 legend). These we call “non-crop” or “outgroup” viruses; many had been isolated from wild plants, although some came from regional specialist crops or indigenous medicinal plants, such as ginseng (*Panax* spp.) and crow-dipper (*Pinellia ternata*; a medicinal arum) (Appendix A).

The countries from which the crop and non-crop isolates came, and the numbers of isolates from each of those countries, are shown in Figure 2. The non-crop isolates came from only 14 countries, most in SE and E Asia and Australia (>80%), whereas the crop virus isolates came from 30 countries worldwide (<50%).

A phylogeny of all 248 n-rec genomic ORFs was used to identify fifteen n-rec sequences from among the non-crop plants with the smallest average patristic distance to each of the seven crop viruses, excluding SbMV and WMV as these had fewer than nine n-rec sequences. We call these sets of “outgroup” viruses as some included sequences from other crop virus clusters, but no more than two sequences were included from any other crop virus set. The Accession numbers, hosts, and provenances of these sequence sets, 232 ORFs in total, are listed in Appendix A, and Figure 3 shows their phylogeny. It can be seen that PeMoV has a unique set of non-crop viruses closest to it, DashMV and ZYMV mostly share another set of non-crop viruses, and the other four crop viruses are closely related and share different combinations drawn from a single cluster of non-crop and crop viruses.

### 3.2. Recombinants

Genetic recombination is often stated to be an important source of genetic diversity in virus population evolution [44,45,46], and recombinants have been found to be common in the populations of many potyviruses. Recombination has been previously reported in populations of all the crop BCMV subgroup viruses [33,47,48,49,50,51,52]. Our RDP analysis of all 731 genomic ORFs of the BCMV subgroup found that only 248 (34%) of the sequences showed no significant evidence of recombination (see Section 2).

If a particular recombination event had been a key trigger establishing a spillover, then one would expect all isolates resulting from that spillover to show evidence of that event, though subsequent recombination events in some isolates might obscure the first event. The RDP analysis showed that most WMV isolates and most SbMV isolates were recombinants; therefore, it is possible that the spillovers of these two viruses were triggered by recombination. These two viruses are closely related crown clusters of a BCMV sub-sub-lineage with isolates of uraria mosaic virus (LC477217) and wisteria vein mosaic virus (NC_007216) forming the basal clusters. The nearest major “parent” of WMV (Figure 4) is a Korean isolate of EAPV (LC656468), and its nearest minor parent is a Chinese isolate of BCMV (MW834586) from poplar (*Populus alba* var *pyramidalis*); the recombinants were produced by Event 60, identified in the RDP analysis of all the ORF sequences (seven methods RGBMCS3, mean *p* < 10^−21^), and were found in 121 isolates. The two most basal nodes of WMV branch from isolates (KF274031, KC845322) isolated from the Chinese hosts *Ailanthus altissima* (‘Tree of Heaven’; Sapindales) and *Atractylodes macrocephala* (a medicinal plant; Asterales). The next most basal node branches from isolates (MK217416 and KX926428) from *Panax ginseng* (a medicinal plant, Apiales) and *Alcea rosea* (hollyhock, Malvales). The third and subsequent nodes subtend clusters of WMV isolates, mostly from cucurbits, and grow worldwide.

The basal events of the SbMV phylogeny (Figure 4) are superficially similar to those of WMV. The basal lineage of SbMV is of sequences from isolates found in Pinellia spp. (a medicinal arum) and are of recombinants closest to DashMV, now found in Typhonium giganteum, the Giant Voodoo Lily, a Chinese ornamental arum. The main SbMV population is of two lineages arising from RDP Event 9 found in 21 sequences and from RDP Event 90. However, there is no SH statistical support distinguishing whether the main SbMV lineages arose before or after the divergence within the Pinellia cluster when the recombinant regions are removed and the n-rec remnants are used in a BLASTn search of GenBank; then, sequences of WMV (LC787269, MT780537) and Calla lily latent virus (EF105298/9) are found to be closest.

None of the other crop viruses show evidence that recombination was associated with the spillover events; there are many recombinants in the BCMV population, but all are sub-lineage specific.

### 3.3. Dating

Analyses using the TempEst v. 1-5-3 and IQ-TREE v.2.3.4 programs found ”temporal signals” only in the EAPV, PeMoV, and ZYMV n-rec ORF individual datasets, and not in the alignment containing all 232 n-rec sequences. The IQ-TREE analyses found that EAPV sequences had a “most recent common ancestor” (MRCA) date of 8717 BCE with its major divergences occurring after 1592 CE, and, likewise, the PeMoV population had a single basal divergence dated 173 CE but with its major divergence occurring after 1894 CE, and the ZYMV population had a MRCA of 683 CE, and all its crown clusters formed after 1500 CE.

Comparing these datings, and perhaps extrapolating them in detail to the other four crop viruses, was unlikely to be useful as the histories of some of the viruses obviously involved significant host changes with concomitant changes in selection. However, in all the phylogenies, the crop virus MRCAs were older than 1500 CE (i.e., pre-Columbian world marine trade), and all of the nodes subtending their major crown clusters were (see Section 4).

### 3.4. Population Genetics

The sequences of the ORFs of seven n-rec crop sequence populations and the corresponding outgroup virus sequences were compared using basic population genetic metrics calculated using the DnaSP program suite, and the results are in Table 1. They could not be calculated for SbMV and WMV, as no n-rec sequences of WMV were found—and only three of SbMV—and a minimum of six sequences are required for calculating most population genetic metrics.

The nucleotide diversities of the different viral datasets showed (Table 1), as expected, that the crop virus sequences were all less variable than their respective outgroup ones; mean crop virus sequence numerically weighted diversity 0.102 ± 0.100 compared with mean outgroup virus sequence diversity 0.283 ± 0.035. PeMoV had the least diverse set of crop sequences, with the smallest nucleotide diversity (π = 0.02) and number of segregating sites (S = 1231), but had the most diverse set of outgroup sequences (π = 0.379, and S = 6201), which reflects the fact that its branches are basal in the phylogeny and its mostly unique group of outgroup sequences were therefore on long branches (Figure 3).

All the outgroup populations gave large positive Tajima’s D estimates that are statistically significant (Table 1), which is evidence of “balancing selection and/or significant population contraction” as “rare alleles are scarce” [53]. Although Tajima’s D estimates for all the crop sequences were more negative than those of the outgroup sequence sets, unexpectedly, only those of PeMoV had a statistically significant negative Tajima D estimate, indicative of a “population expansion after a bottleneck, or a recent selective sweep” resulting in “an excess of rare alleles”, namely a spillover. Only 15 full-length n-rec genomic sequences of PeMoV were in GenBank when this project started in 2023, and they formed a single crown cluster, although the sequences came from isolates collected in five countries (Brazil, China, Iran, Korea, and Turkey). However, although the populations of the other crop viruses had several crown clusters mixed with basal outliers, samples of isolated crown clusters of each of the nine to fifteen sequences of those viruses did not give statistically significant negative Tajima D values; these crown clusters were mostly of the ORFs of isolates collected from single countries. Estimates for individual genes found that the P1, HC-pro, and P3 genes had larger Tajima’s D values than the others, suggesting that they are more responsive to selection changes, and the PIPO gene had consistently smaller Tajima’s D values than the other genes, suggesting that it had been less affected by selection.

The dN/dS estimates showed that all the sequences from crop isolates, except those of DashMV, were under greater negative selection (i.e., smaller dN/dS) than those from the corresponding outgroup sequences, properly reflecting increased purifying selection. Furthermore, dN/dS scans (Figure 5 for BCMV crop) showed that negatively selected codons are present throughout the genome, whereas positively selected codons are clustered in the P3 gene, which encodes two proteins, P3 and P3N-PIPO, reported to function as movement proteins [23,54,55].

As described above, the potyvirus P3 gene (codons 1141 to 1504) has a motif, “5-GAAAAAA-3” in its centre, which causes slippage when being transcribed [20], and as a result, a small proportion of the progeny transcripts [56] of the P3 gene have an extra “A” in the slippage region and translate the C-terminal region of the P3 gene in its + 1 reading frame. Thus, the resulting protein, P3N-PIPO has a P3 N-terminal region and a different C-terminal region, which has been named PIPO (“Pretty Important Protein Overlapping”). The P3 gene is 364 codons long, and the slippage motif is 165 codons from its 5′-terminus; however, the PIPO region is only 71 to 94 codons long in different viruses, and thus, the P3N-PIPO protein is smaller than the P3 protein.

We therefore further investigated the number and position of individual positively and negatively selected codons in the ORFs and P3N-PIPO regions of the different crop and outgroup datasets using FUBAR (Fast Unconstrained Bayesian AppRoximation for inferring selection) (Appendix A and Appendix A), and Figure 6 summarises the results of those analyses.

The FUBAR analyses (Figure 6) agree with the dN/dS results in finding that most codons were under negative selection in all sequence comparisons. The crop and outgroup ORFs had similar numbers of positively selected codons (1.43 codons/genome (±1.39) in the crop virus ORFs compared with 1.86 (±0.90) in outgroup virus ORFs); however, the distribution of those sites throughout the ORFs was not random, and 80% were in the P3 gene (codons 1141–1504), in agreement with the dN/dS scan results (Figure 4). The P3 codons that were positively selected most frequently (Figure 5) were codons 1316 and 1326 (five sets each) and codon 1336 (four sets). All but two of the positively selected codons were in the PIPO encoding region of P3. Furthermore, in the P3N-PIPO sequences, codons 1366 (five comparisons) and 1375 (four comparisons) were the codons positively selected most frequently. Only three pairs of positively selected codons overlapped in the PIPO region: codon 1340 in P3 with codon 1339 in P3N-PIPO, codon 1340 in P3 with codon1340 in P3N-PIPO, and codon 1347 in P3 with codon 1346 in P3N-PIPO.

Finally, to better understand the results of the FUBAR analyses, we modelled the potyvirus proteins involved using the online AlphaFold3 server [42]. We used proteins of two of the viruses of the BCMV subgroup: PeMoV (sequences KF977830 and MZ442685) and DashMV (KT026108 and KY242358). These were chosen because patristic distance comparisons of the nucleotide and encoded amino acid sequences of all 232 n-rec sequences showed that those of PeMoV and DashMV were the most different, and hence most likely to separate functional and structural similarities and differences from biological variation.

AlphaFold3 models of the P3 and P3N-PIPO amino acid sequences (Figure 7) show what complex and interesting molecules they are. The P3 and the P3N-PIPO proteins have identical N-terminal regions. These consist of a series of eight short alpha-helices folded to form an irregular polyhedral head. The N-terminal regions of the P3 C-terminal and PIPO regions form “brims” of longer helices around the heads of both proteins. The brim in P3 includes the positively selected codons (1316, 1326, and 1336). The C-terminal regions are significantly different in structure and composition; the first part of the P3 protein is a long helix of 106 amino acids with a C-terminus of two short helices that fold to lie across the long helix, whereas the PIPO protein, which is shorter, has a helix of 53 amino acids with a bend after the 35th, and a C-terminal region of 20 amino acids that are “intrinsically disordered”.

The isoelectric points (pIs) estimated by ProtParam of the P3N head proteins are, on average, 6.4, but the pIs of the brim helices, the long helix of P3, and the bent helices of PIPO are from 9.5 to 10.3; however, the P3 C-terminal short helices have pIs of 4.45.

The hydropathicity indices of the long P3 helix and folded PIPO helices are positive, indicating that they are hydrophobic, whereas all other parts of P3 and P3N-PIPO are negative and hence likely to be hydrophilic. The PIPO protein is compositionally less variable than the regions of the P3 protein, and all three regions (i.e., the helices and the ID C-terminus) have pIs around 10.

The elongated shape of the P3 proteins is intriguing. The fact that the long helix providing that shape is of 106 amino acids in all 232 n-rec sequences, and all indels in the P3 gene are confined to regions of the N and C-termini that have no effect on the length of the protein, suggests that the length has functional significance. The acidic composition of the C-terminus compared with the helix and head to which it is attached again suggests functional significance. The length of alpha-helical peptides is determined by the backbone: 3.6 residues/turn with each turn adding 0.54 nm along the axis. Therefore, the P3 long helix of 106 amino acids is 15.9 nm long, and as it constitutes about 80% of the length of the protein, the long axis of the P3 protein is about 20.5 nm and its width maximally 5 nm. This suggests that the P3 protein would not pass from cell to cell through unaltered plasmodesmata, although this is a complex issue.

AlphaFold3 will also co-model (i.e., model and dock) two or more proteins. We therefore made a search of all combinations of the P3 and P3N-PIPO proteins of PeMoV (KF977830) with each of its other nine proteins and found the P3 and CI protein combination gave the largest “interface temperature modelling score” (ipTM) of 0.6 with ipTMs of >0.5 being considered a ”true structure” and the other combinations giving an average ipTM of 0.15. To check whether this was biologically informative, we co-modelled all combinations of the P3 and CI molecules of two isolates of PeMoV and DashMV: KF977830 with KT026108 and MZ442685 with KY242358. The models gave varying PTMs and ipTMs around the values above, but those with the largest ipTM consistently showed the brim region (codons 1279–1286) of the head of the P3 protein closest to a region of the CI protein where there was a helix (codons 1814–1825) and a short anti-parallel beta-strand (codons 1830–1834). Thus, none of these regions defined structurally involved the codons found by FUBAR to be most frequently positively selected.

## 4. Discussion

Our study, using full-length genomes of a large number of BCMV subgroup viruses, has confirmed an earlier analysis of their CP gene sequences [17], which concluded that the BCMV subgroup of potyviruses probably first diverged in uncultivated plants (or local specialist crops) in SE and East Asia and spread as far as Australia, and more recently, it also produced some major crop pathogenic viruses that have spread worldwide. Here, we used phylogenetic and population genetic methods to compare the crop viruses with the most closely phylogenetically related non-crop or outgroup viruses looking for differences that might have occurred during each of nine spillovers from a non-crop virus to a major crop pathogen.

We have identified nine viruses of the BCMV subgroup of potyviruses as major pandemic viruses. Three of them (DashMV, CpAbMV, and SbMV) were isolated from dasheen or taro, cowpea, and soybean, which are among the earliest plants to be domesticated and have been grown for many thousands of years in SE Asia. Cowpea (*Vigna unguiculata*) was domesticated in sub-Saharan Africa before 2500 BCE and by 400 BCE was long established in all of its modern major production regions of the Old World, including Africa, the Mediterranean, India, and Southeast Asia [58]. Taro (*Colocasia esculenta*) is a member of the *Araceae* and probably the earliest domesticated crop. Archaeological studies indicate that the crop has been cultivated for at least 28,000 years [59] over a vast area spanning Africa and India to South China, Melanesia, and northern Australia [60]. Taro is polymorphic and the shape of the corm distinguishes dasheen (*Colocasia esculenta var. esculenta*) and eddoe (*Colocasia esculenta var. antiquorum*) types [61], and genetic analyses indicate that taro may have been domesticated on multiple occasions. Soybean (*Soja max*) was domesticated in East Asia more than 3000 years ago from wild *Glycine soja* and other Glycine species found in the eastern regions of the Old World and Oceania. Interestingly the phylogenies of n-rec DashMV and CpAbMV (Appendix A) are distinct from those of the other pandemic viruses in the subgroup by consisting of a few long-branch lineages; there were no n-rec SbMV sequences in our data. A fourth virus, WMV, probably moved from various uncultivated plants and minor crops and became a crop pathogen in NE Asia when watermelon (*Citrullus lanatus*) became widely grown there around 1000 years ago [33]. The other five viruses were isolated from crops domesticated in the Americas [62]: two from common bean (*Phaseolus vulgaris*) and one each from passion fruit (*Passiflora edulis*), peanut (*Arachis hypogeae*), and zucchini (*Cucurbita pepo*). Thus, several of the nine pandemic viruses of the BCMV subgroup were generated from “new encounters” of the sort described and discussed by Refs. [1,63], and it is likely that the spread of the host crop in worldwide trade and to new agricultural areas is the main factor in them having become major crop pathogens. This was confirmed in our limited dating analyses, which showed that although many of the crop viruses had “Most Recent Common Ancestors” back to 8700 BCE (EAPV), none of the nodes subtending their major crown clusters were older than 1500 CE, which was when the post-Columbian era of world marine trade started [64,65]. However, there is also the possibility that this dating has been influenced by the “time dependent rate phenomenon” [66].

It is the carriage by seafarers of infected plant material, especially seeds, that most likely explains the presence of BCMV subgroup viruses on at least three oceanic islands (Figure 2) and the “genetic connectivity” of ZYMV isolates in SE Asia and NW Australia [67,68]. These viruses are less likely to have been spread long distances over oceans by aphids as potyviruses do not persist in flying or starving aphids. “The plant virus transmissions database” [69] records that seven of the nine crop viruses we have studied are “seed-borne”, but that DashMV and WMV are “not seed-borne”, and although that may be correct for DashMV as the host is vegetatively propagated, it is questionable for WMV given its worldwide distribution; whether or not a virus is recorded as seed-borne depends greatly on the number of seeds that were tested.

Genetic recombination may represent a significant evolutionary force for plant RNA viruses [70,71,72]; for example the necrogenic lineages of potato virus Y are mostly recombinants of non-necrogenic lineages [73]. However, there is no evidence that recombination provides new crop species of viruses; indeed, Moreno et al. [74] studied a WMV population in melon with 7% recombinants and found “strong selection against isolates with recombinant proteins, even when originated from closely related strains.” Desbiez and Lecoq [75] confirmed that “the P1 of WMV was 135 amino-acids longer than that of SMV, and the N-terminal half of the P1 showed no relation to SMV but was 85% identical to BCMV. This suggests that WMV has emerged through an ancestral recombination event”. However, we find that this is not supported by the more detailed recombination events now revealed; the large proportion of recombinants found in WMV and SbMV populations may not only indicate the sharing of infected seeds but also reflect the larger number of genomes that have been compared, as this makes it more likely for recombinants to be found. None of the other seven BCMV subgroup viruses showed evidence of recombination being involved in their emergence.

Like many studies of potyviruses already published, we have found evidence of strong negative selection in all nine BCMV subgroup viruses. Our arbitrary method for distinguishing crop and non-crop viruses seems to be effective as they gave different, but consistent, datasets; for example, the non-crop viruses were consistently under more negative selection than the crop viruses, and the populations of their hosts were declining. The finding that most of the positively selected codons of these viruses were concentrated in their P3 proteins, and the P3N-PIPO proteins derived from them, was significant, especially as more were found in the analyses of the non-crop viruses than the crop viruses. This suggests that the selection identified by FUBAR is linked in some way with host preferences as the hosts of the outgroup viruses were more taxonomically diverse than the crop virus hosts (Appendix A); the crop viruses were isolated from plants of nine families (*Araceae*, *Berberidaceae*, *Fabaceae*, *Moraceae*, *Orchidaceae*, *Passifloraceae*, and *Pedaliaceae*), and the outgroup viruses from plants of 13 families (*Amaranthaceae*, *Apocynaceae*, *Asparagaceae*, *Asphodelaceae*, *Basellaceae*, *Begoniaceae*, *Fabaceae*, *Hyacinthaceae*, *Iridaceae*, *Liliaceae*, *Melanthiaceae*, *Orchidaceae*, and *Passifloraceae*), but with only three families shared (*Fabaceae*, *Orchidaceae*, and *Passifloraceae*). However, no correlations were found between phylogenies of the nucleotide or encoded amino acid sequences of the P3 or P3-PIPO genes or the positively selected codons found in them, and the host differences (i.e., monocotyledonous versus dicotyledonous hosts; asterids versus rosids versus caryophyllid hosts [76]).

AlphaFold3 models of the P3 and P3N-PIPO amino acid sequences show what complex and interesting molecules they are. The elongated shape of the P3 proteins is intriguing, and the fact that the long helix providing that shape is of 106 amino acids in all 232 n-rec sequences despite large sequence differences suggests that the length has functional significance and the protein may be membrane-spanning. The acidic composition of the C-terminus compared with the helix and head to which it is attached again suggests functional significance. As potyviruses cause such significant damage in many crops worldwide, much international effort is currently being made to understand the molecular biology of their replication as, for example, a knowledge of how they spread from cell to cell with the plant might provide methods to control that process. However, there are significant differences of opinion about which 11 potyvirus proteins are involved [23,77,78,79]. The acidic composition of the C-terminus compared with the helix and head to which it is attached, as well as the size and chemistry of the various regions of P3, again suggests functional significance.

## Figures and Tables

**Figure 1 viruses-16-01351-f001:**
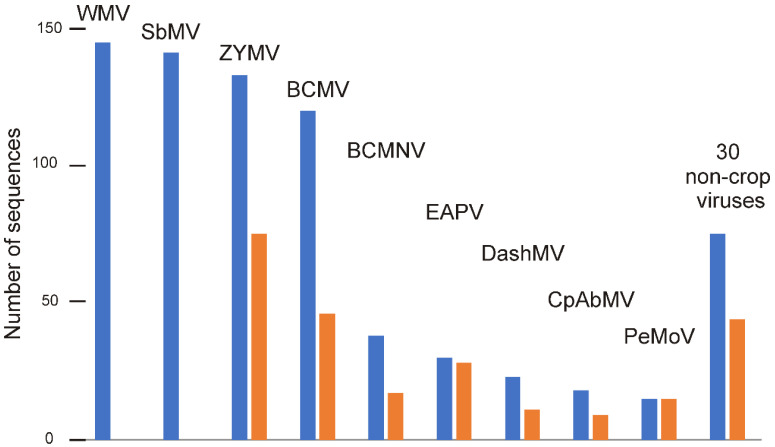
A histogram showing the numbers of sequences of different BCMV subgroup viruses in GenBank (blue bars) found in crops, and the numbers of those sequences found to be non-recombinant (n-rec; orange bars). Virus acronyms: WMV, watermelon mosaic virus; SbMV, soybean mosaic virus; ZYMV, zucchini yellow mosaic virus; BCMV, bean common mosaic virus; BCMNV, bean common mosaic necrosis virus; EAPV, East Asian passiflora virus; DashMV, dasheen mosaic virus; CpAbMV, cowpea aphid-borne mosaic virus; PeMoV, peanut mottle virus. The histogram also shows the total and n-rec numbers of sequences of non-crop BCMV subgroup viruses; they were eight sequences of hardenbergia mosaic virus (seven n-rec); six of telosma mosaic virus (one n-rec); four of basella rugose mosaic virus (four n-rec), Paris mosaic necrosis virus (three n-rec), and wisteria vein mosaic virus (two n-rec); three each of beet mosaic virus (two n-rec), freesia mosaic virus (two n-rec), passionfruit Vietnam virus, and passionfruit woodiness virus (three n-rec); two each of begonia flower breaking virus (two n-rec), blue squill virus A (two n-rec), calla lily latent virus, passiflora chlorosis virus (two n-rec), passiflora foetida virus Y (two n-rec), yam bean mosaic virus (one n-rec), and zantedeschia mild mosaic virus (two n-rec); and one each of achyranthes bidentata mosaic virus (one n-rec), atractyloides macrocephala virus, fritillary virus Y (one n-rec), gomphocarpus mosaic virus (one n-rec), impatiens flower break potyvirus (one n-rec), keunjorong mosaic virus (one n-rec), passiflora virus Y (one n-rec), passionfruit severe mottle virus, pleione flower breaking virus (one n-rec), polygonatum kingianum virus 3 (one n-rec), polygonatum kingianum virus 4 (one n-rec), polygonatum mosaic-associated virus 1 (one n-rec), saffron latent virus isolate (one n-rec), and uraria mosaic virus (one n-rec). Sequences of blackeye cowpea mosaic virus, lygodium japonicum potyvirus, peanut stripe virus, and poplar mosaic virus (China) were phylogenetically indistinguishable from those of bean common mosaic virus and were pooled with them and that of vanilla mosaic virus with dasheen mosaic virus. Likewise, those of artemisia carvifolia potyvirus, cerasus yedoensis potyvirus, and cucurbita moschata potyvirus were pooled with those of watermelon mosaic virus, soybean virus A with those of wisteria vein mosaic virus, and of allium fistulosum potyvirus, brassica caulorapa potyvirus, cerasus yedoensis potyvirus, cucurbita moschata potyvirus, luffa aegyptiaca potyvirus, sapindus mukorossi potyvirus, and solanum melongena potyvirus with those of zucchini yellow mosaic virus.

**Figure 2 viruses-16-01351-f002:**
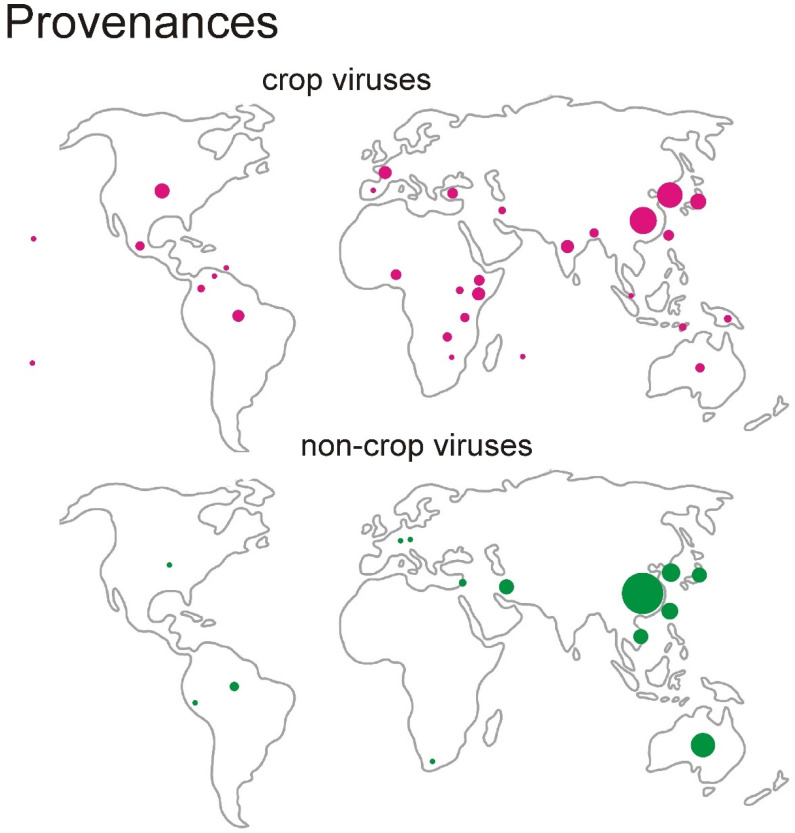
A diagram showing the 30 countries (upper map and red discs) from which the complete ORFs of BCMV subgroup crop viruses have been isolated, and the corresponding 14 countries (lower map and green discs) providing the ORFs of non-crop virus isolates. The disc sizes are related to the number of isolates obtained from each country and range from 61 non-crop virus sequences from China to single isolates from three oceanic islands (Cook, Hawaii, and Reunion); the diameter of each disc is scaled to the square root of the number of isolates from that country.

**Figure 3 viruses-16-01351-f003:**
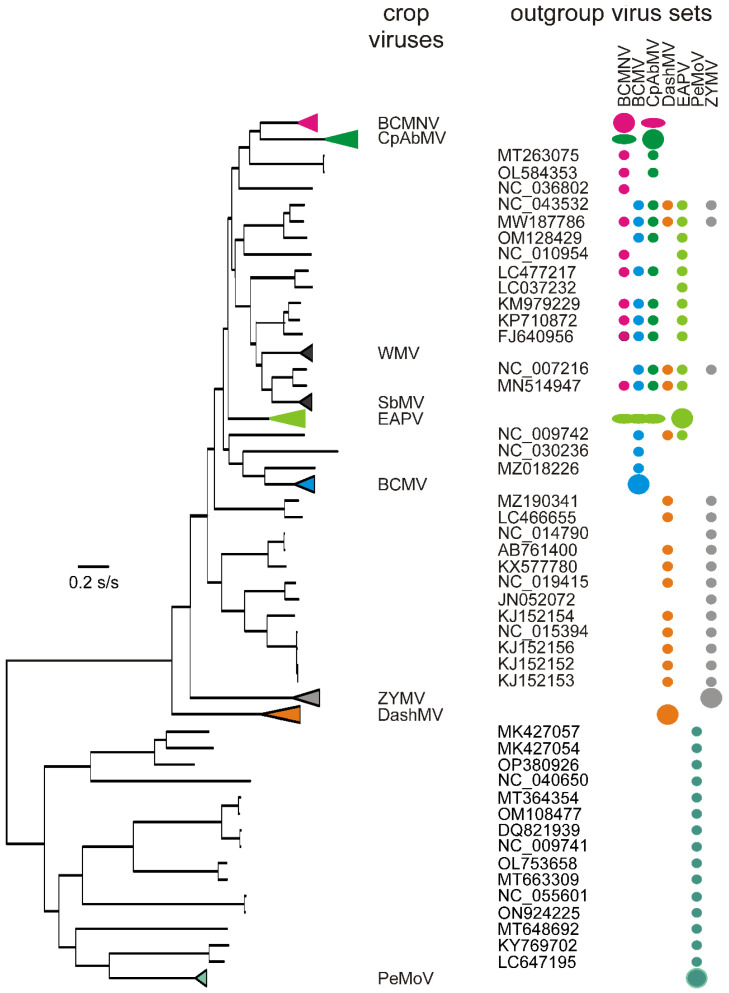
A phylogeny of the seven n-rec crop virus sequence sets (Appendix A) and the 45 non-crop isolates closest to them. The crop virus clusters have been collapsed and represented as triangles, and these, together with the large and small discs, are colour-coded to indicate the groupings used for population genetic comparisons. There were too few n-rec SbMV sequences for useful population genetic comparisons, and no n-rec WMV sequences, so these two viruses are represented in Figure 3 by black triangles. Crop virus sets contributing outgroup sequences to other crop sequence sets are marked with coloured ellipses.

**Figure 4 viruses-16-01351-f004:**
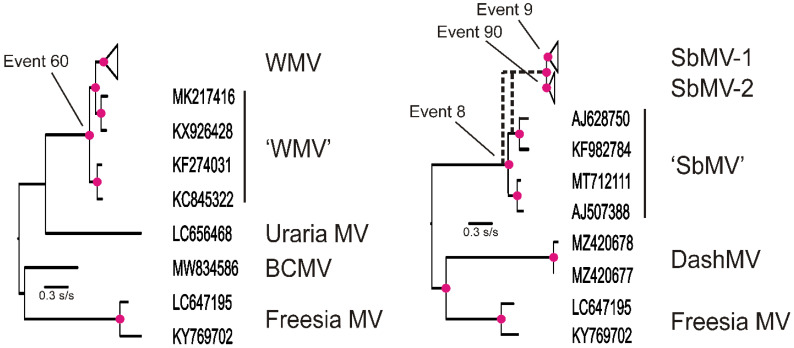
Basal recombinants. Phylogenies showing the relationships between the sequences basal to the WMV and SbMV sequences from crop isolates, which are collapsed and represented as triangles. Four WMV isolates involved in Event 60 of the RDP analysis are labelled “SbMV”, and were obtained from unusual hosts in NE Asia, whereas the relationships of the four SbMV isolates from Pinellia spp. are more complex and unresolved statistically. Nodes marked with a red disc have >0.99 SH statistical support. The results of the RDP5 analyses are available from the authors upon request.

**Figure 5 viruses-16-01351-f005:**
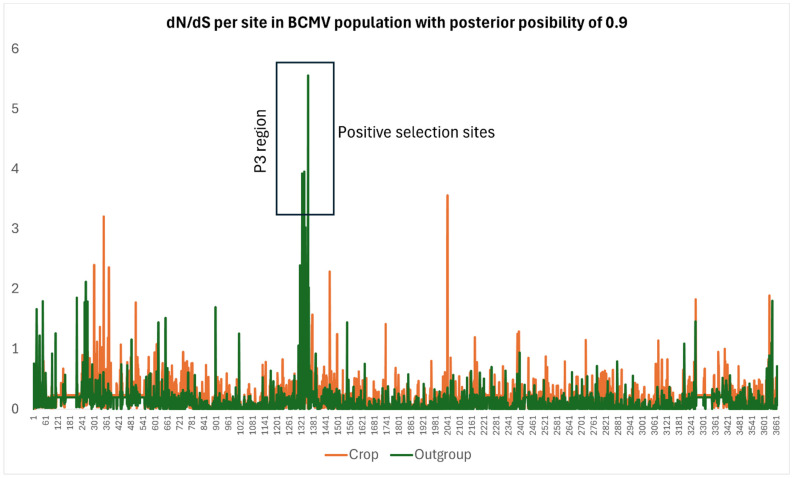
dN/dS map of the 60 main ORFs of the BCMV genomes obtained using the online Datamonkey server (https://www.datamonkey.org/ (accessed on 30 November 2023)) and drawn by Excel.

**Figure 6 viruses-16-01351-f006:**
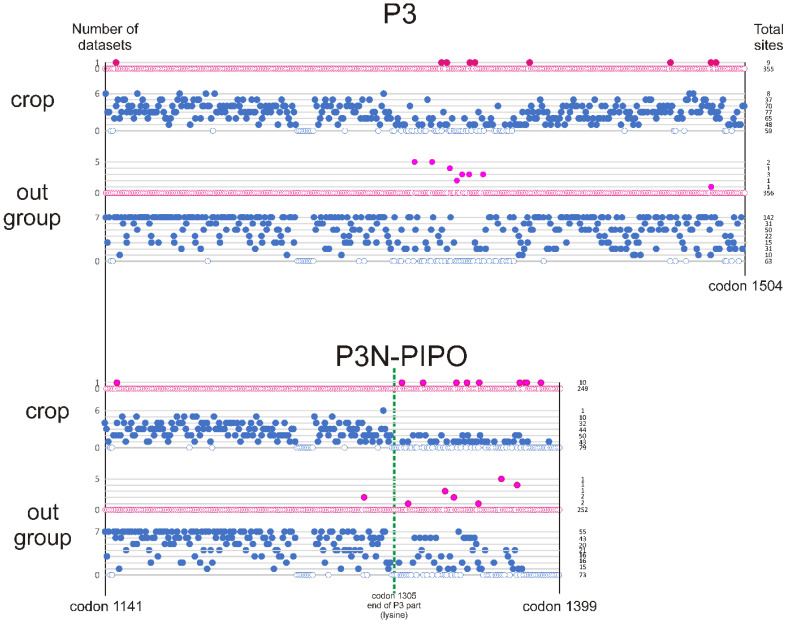
Summary of the FUBAR analyses for positively and negatively selected sites in the P3 and P3N-PIPO genes of seven viruses and their outgroups (Appendix A); the same site was never found to be positively selected in both the virus dataset and its outgroup. Positively selected sites are red discs; negative are blue. Discs for zero positive or negative sites are shown as circles; individual discs show how many viruses (0–7) gave a significant positive or negative FUBAR result. Rows separate the numbers of datasets (0–7) providing each point; total sites/row at right. Green dotted line shows the position of the 3′most codon of the P3N gene. Codon numbering from the 731-sequence alignment used in this study.

**Figure 7 viruses-16-01351-f007:**
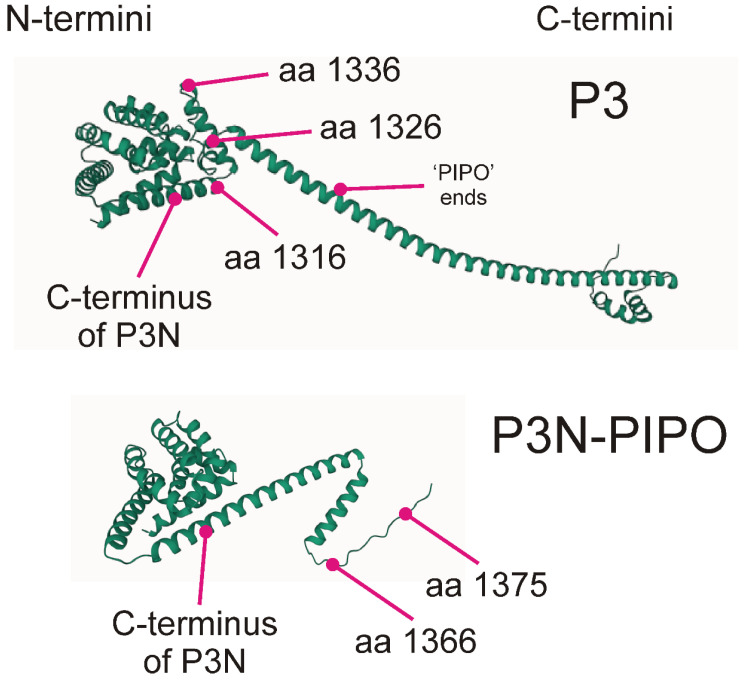
AlphaFold3 models of the P3 and P3N-PIPO proteins encoded in the genome of PeMoV (KF977830). Support for the models of P3 and P3N-PIPO proteins of several BCMV subgroup viruses is judged by the “predicted temperature modelling score” (pTM), which was all around 0.6, where above 0.5 means the overall model “might be similar to the true structure” [57], especially as all had essentially the same structure.

**Table 1 viruses-16-01351-t001:** The population genetic parameters and demography tests estimated for the alignments of seven crop viruses, their outgroups, and their combined alignments.

Virus	Number of Sequences	Nucleotide Diversity	Number of Segregating Sites	Tajima’s D	dN/dS
BCMNV-all	32	0.244	5529	2.297 *	0.221
BCMNV-crop	17	0.069	1803	0.674 ns	0.089
BCMNV-outgroup	15	0.299	5472	2.678 **	0.237
BCMV-all	60	0.194	5689	1.593 ns	0.185
BCMV-crop	45	0.114	4499	0.104 ns	0.106
BCMV-outgroup	15	0.287	5419	2.501 **	0.221
CpAbMV-all	23	0.294	5455	3.252 ***	0.233
CpAbMV-crop	8	0.212	4311	1.092 ns	0.137
CpAbMV-outgroup	15	0.290	5344	2.616 **	0.229
DashMV-all	26	0.311	5780	3.442 ***	0.261
DashMV-crop	11	0.306	5386	2.516 **	0.264
DashMV-outgroup	15	0.311	5623	2.807 **	0.261
EAPV-all	43	0.195	5378	1.553 ns	0.217
EAPV-crop	28	0.054	2880	−1.155 ns	0.163
EAPV- outgroup	15	0.269	5147	2.387 *	0.199
PeMoV-all	30	0.304	6248	2.632 **	0.362
PeMoV-crop	15	0.020	1231	−2.273 **	0.081
PeMoV-outgroup	15	0.379	6201	3.273 ***	0.370
ZYMV-all	90	0.176	5727	1.371 ns	0.189
ZYMV-crop	75	0.095	3564	0.669 ns	0.072
ZYMV-outgroup	15	0.286	5315	2.624 **	0.238

ns = not statistically significant; * = significant at *p* < 0.1; ** = significant at *p* < 0.01; *** = significant at *p* < 0.001.

## Data Availability

All data generated or analysed during this study are included in this published article and its Appendix A. Further details are available from the corresponding authors upon reasonable request.

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
