# Peer review of "A Genetic Study of Spillovers in the Bean Common Mosaic Subgroup of Potyviruses"

_viruses, 2024, doi:10.3390/v16091351_

Round 1

Reviewer 1 Report

Comments and Suggestions for Authors

This article is based on an in-depth knowledge of the ecology, evolution and history of the bean common mosaic virus (BCMV) potyvirus subgroup and of their hosts, and on the information extracted from a large full-length sequence dataset by using a suite of phylogenetic, population genetics and 3D modeling analyses. Complementary and/or alternative methods may have been applied but are unlikely to bring additional or different results. The number of sequences in Genbank was used to sort the BCMV subgroup sequences into the “crop” and “non-crop” or “outgroup” viruses « because crop pathogens are of more interest to plant pathologists than non-crop pathogens » (over 700 in total). Although somewhat arbitrary, as acknowledged and discussed by the authors, this distinction is justified a posteriori by the consistent differences of properties found between the two groups. The article is easy to follow and the results obtained are interesting. Some of them confirmed earlier results, others provide new ideas on the features that permit or promote adaptation and/or spillovers, for instance that positively selected codons are involved in host range determination, and on the role of the long alpha helix of the P3 protein.

I have listed below a few remarks and queries

Spillover

- Is there evidence that some of the lineages of the BCMV subgroup are more prone than others to generate spill-overs on crops (figure 3)?

- The finding that only PeMoV had a statistically significant negative Tajima estimate, indicative of a “population expansion after a bottleneck, or a recent selective sweep” resulting in “an excess of rare alleles”, namely a spillover (lines 307-308) is unexpected. Is it a limitation of the dataset and/or of the method used ?

dating

- The dating part is not sufficiently documented to be assessed (no figures and no data given).

- A positive Tempest linear relationship is a necessary but not the sufficient condition of a temporal signal.

- The possible impact of Time Dependent Rate Phenomenon on the dating is not considered to warrant the split before and after 1492.

Recombination

- The RDP criteria used in this study are rather conservative of the non-recombination  hypothesis as only genome portion detected by more than four methods with a very low probability level in RDP are considered as recombinants.

- "Only one third of crop virus genomes are non-recombinant, but more than half the non-crop virus genomes are". Can we exclude that with an even higher sequencing effort, even less sequences would be found non-recombinant? If so, could it impact the conclusions?

Others

- The lower Dn/Ds values in the crop viruses are not discussed.

- Line 485 « the non-crop viruses were consistently under more negative selection than the crop viruses ». On the contrary, Table 1 indicates that crop viruses are under more negative selection than the crop viruses.

- Did the recent metagenomics programs greatly enlarge the number of sequences of non-crops BCMV viruses ?

Editing points

- Lines 30-31 : omit « (List three to ten pertinent keywords specific to the article yet reasonably common within the subject discipline.)

- Lines 151-152 : omit « This section may be divided by subheadings. It should provide a concise and precise description of the experimental results, their interpretation, as well as the experimental conclusions that can be drawn ».

- Line 254, corrrect « paratheses ».

- Line 256, written « in parentheses », is it rather « in brackets » ?

- Line 464 « as potyviruses do not persistent in flying or starving aphids ». rather «  … do not persist in... » ?

Author Response

We thank the Referees and Editors for their comments (knowledgeable and positive), and have responded to all of them in the new versions of the paper and Supplementary Data. 

Spillover

  • Is there evidence that some of the lineages of the BCMV subgroup are more prone than others to generate spill-overs on crops (figure 3)?

Response: No evidence from this study – too small.

  • The finding that only PeMoV had a statistically significant negative Tajima estimate, indicative of a “population expansion after a bottleneck, or a recent selective sweep” resulting in “an excess of rare alleles”, namely a spillover (lines 307-308) is unexpected. Is it a limitation of the dataset and/or of the method used ?  

Response: We have added a comment explaining that our Tajima’s D check of ‘crown clusters’ of similar appearance to the PeMoV cluster was ineffective as the ‘crown clusters’ available from other crop clusters were not comparable as most came from single countries.

dating

  • The possible impact of Time Dependent Rate Phenomenon on the dating is not considered to warrant the split before and after 1492.  

Response: Forgot to include that possibility.  Have now included comment and the Ho et al reference.

Recombination

  • The RDP criteria used in this study are rather conservative of the non-recombination  hypothesis as only genome portion detected by more than four methods with a very low probability level in RDP are considered as recombinants.  

Response: True.

  • "Only one third of crop virus genomes are non-recombinant, but more than half the non-crop virus genomes are". Can we exclude that with an even higher sequencing effort, even less sequences would be found non-recombinant? If so, could it impact the conclusions?  

Response: Absolutely correct so a line covering that is included.  We did not report that RDP analyses of random subsamples of the WMV sequences, decreased the proportion of recombinants found – the smaller the sample, the smaller the apparent recombinant rate!!!

Others

  • The lower Dn/Ds values in the crop viruses are not discussed.

Response: A sentence added (lines 363-364).

  • Line 485 « the non-crop viruses were consistently under more negative selection than the crop viruses ». On the contrary, Table 1 indicates that crop viruses are under more negative selection than the crop viruses.

Response: Apologies – many thanks for finding that mistake.

  • Did the recent metagenomics programs greatly enlarge the number of sequences of non-crops BCMV viruses ?  

Response: No, but not specifically checked.  An earlier check of all metagenomic potyvirids showed that many were “mosquito-associated” or “slug-associated” but clearly plant potyvirids, and one was an in-silico recombinant with its swop-over point in the middle of the GDD-encoding region of the RdRp, and the separated parts were very close to two separate plant potys!!!

Editing points

  • Lines 30-31 : omit « (List three to ten pertinent keywords specific to the article yet reasonably common within the subject discipline.)  

Response: Done

  • Lines 151-152 : omit « This section may be divided by subheadings. It should provide a concise and precise description of the experimental results, their interpretation, as well as the experimental conclusions that can be drawn ».

Response: Done

- Line 254, corrrect « paratheses ».

  • Line 256, written « in parentheses », is it rather « in brackets » ?

Response: Thanks.  I now know the meaning o ‘in parenthesis’.

  • Line 464 « as potyviruses do not persistent in flying or starving aphids ». rather «  … do not persist in... » ? 

Response: Done.  thanks

Reviewer 2 Report

Comments and Suggestions for Authors

Section “Keywords”: Unfortunately, the section is empty.

Section “Results”, Lines 151-153: Obviously, the text fragment “This section may be divided by subheadings. It should provide a concise and precise description of the experimental results, their interpretation, as well as the experimental conclusions that can be drawn” should be deleted.

Section 3.2 and Figure 4: The authors mention several recombination events (named as 8, 9, 60 and 90), but it is not clear from the text where the branch points are located and which parts of the genomes are affected.

Editorial notes:

Lines 20-21: Please, check the meaning of the sentence “As expected, the non-crop viruses are genetically more diverse than the non-crop viruses”.

Lines 78-79: I guess it’s better to replace “conserved polymerase motif of six adenine residues” with “conserved motif of six adenine residues” or “conserved polymerase slippage motif of six adenine residues”.

Lines 288, 318, 336: Please, change the font for “sequences”, “estimates” and “therefore”, respectively.

Lines 444, 456, 460, 497: Extra space(s) between the sentences.   

Author Response

We thank the Referees and Editors for their comments (knowledgeable and positive) and have responded to all of them in the new versions of the paper and Supplementary Data.

Section “Keywords”: Unfortunately, the section is empty. 

Response: Thanks.  Done

Section “Results”, Lines 151-153: Obviously, the text fragment “This section may be divided by subheadings. It should provide a concise and precise description of the experimental results, their interpretation, as well as the experimental conclusions that can be drawn” should be deleted. 

Response: Thanks.  Removed

Section 3.2 and Figure 4: The authors mention several recombination events (named as 8, 9, 60 and 90), but it is not clear from the text where the branch points are located and which parts of the genomes are affected. 

Response: The details are best seen in the original RDP output, so we have reemphasized that we are happy to supply that – on request

Editorial notes:

Lines 20-21: Please, check the meaning of the sentence “As expected, the non-crop viruses are genetically more diverse than the non-crop viruses”. 

Response: Corrected.  Thanks

Lines 78-79: I guess it’s better to replace “conserved polymerase motif of six adenine residues” with “conserved motif of six adenine residues” or “conserved polymerase slippage motif of six adenine residues”. 

Response: Corrected.  Thanks

Lines 288, 318, 336: Please, change the font for “sequences”, “estimates” and “therefore”, respectively.  Response: Done

Lines 444, 456, 460, 497: Extra space(s) between the sentences.   

Response: Removed.